# Microplastic Contamination of Chicken Meat and Fish through Plastic Cutting Boards

**DOI:** 10.3390/ijerph192013442

**Published:** 2022-10-18

**Authors:** Rana Zeeshan Habib, Ruwaya Al Kindi, Feras Al Salem, Wajeeh Faris Kittaneh, Vijo Poulose, Syed Haris Iftikhar, Abdel-Hamid Ismail Mourad, Thies Thiemann

**Affiliations:** 1Department of Biology, United Arab Emirates University, Al Ain P.O. Box 15551, United Arab Emirates; 2Department of Geology, United Arab Emirates University, Al Ain P.O. Box 15551, United Arab Emirates; 3Department of Chemistry, College of Science, United Arab Emirates University, Al Ain P.O. Box 15551, United Arab Emirates; 4Department of Mechanical and Aerospace Engineering, College of Engineering, United Arab Emirates University, Al Ain P.O. Box 15551, United Arab Emirates

**Keywords:** microplastics, plastic cutting boards, food preparation, food contamination, wastewater

## Abstract

Microplastic contamination was found in fish and chicken bought on the market, in food stores and in chain supermarkets in the Middle East with the contamination ranging from 0.03 ± 0.04 to 1.19 ± 0.72 particles per gram of meat in chicken and from 0.014 ± 0.024 to 2.6 ± 2.8 particles per gram in fish. Only one fish was found to be free of microplastic. The source of the microplastic was established to be the polythene-based plastic cutting board the food was cut on. More microplastic contamination was found in food cut from the bone than in cut fillets when the fillets themselves were prepared on surfaces other than plastic. Washing the fish and chicken before food preparation decreased but did not completely remove the microplastic contamination. The fate of the microplastic in grilled fish was studied. The mechanical properties of typical plastic cutting boards commercially used in the markets were investigated in the form of tensile, hardness, and wear tests. Overall, the plastic cutting boards showed similar wear rates.

## 1. Introduction

Microplastic (MP) has been found in different foods, including fish [1,2], crabs [3,4], mussels [5,6] fruit and vegetables [7]. The sources of MPs in these cases are diverse [8], however, often the food sources come into contact with the microplastic in the environment, which is before the actual processing of the food. Typical examples would be the ingestion of microplastics by fish [9,10] or the exposure of vegetables to microplastics on agricultural land [11]. When fish ingest microplastics, the microplastics, if remaining in the fish, mostly reside in the stomach/intestines, which, especially in the case of larger fish, will be cut away, and for the most part will not constitute food for human consumption. Nevertheless, even in those cases, fish offal can be used in fish sauces and in fish meal where microplastic has already been detected [12]. Furthermore, improper gutting in canneries can lead to microplastic contamination of fish [13]. Packaging material is also a source of microplastic/nanoplastic (NP) as is post-food-processing. Micro- and nanoplastic contamination from packaging/bottling has especially been studied in beverages, where the micro-/nanoplastic content in mineral and other bottled water [14,15], fruit juice [16], energy drinks [16] and cold tea [16] has been found to stem from plastic bottles. Plastic tea bags have been found to release micro/nanoplastics upon contact with hot water [17]. Moreover, solid foods such as meat and rice have been known to be contaminated with microplastics from packaging [18,19,20] and single-use plastic containers [21,22,23,24,25].

What effect the ingested microplastic has on human health is still far from completely understood [26]. What is known, however, is that microplastic has been found in human stool [27,28], including in newborn babies [29]. Nanoplastics have been found in human blood and while it has been shown experimentally that microplastics of less than 5 μm can migrate to the liver in fish [30], it has been realized that microplastic can also migrate to other organs, including into muscle tissue [31]. In human beings, it has been found in different organs including the placenta [29,32]. Moreover, it has been shown that microplastic particles of 0.1 μm or less in size can enter hepatocytes from circulation and result in liver damage even at low concentrations [33]. Bonanomi et al. found that carboxylate—modified polystyrene nanoparticles with an average size of 0.5 μm were internalized in CCD-18 human colon cells and led to metabolic changes [34].

Recently, we published our finding that the commercial processing of goat meat and beef on plastic cutting boards leads to microplastic contamination of the meat [35]. Meanwhile, Dawson et al. have reported on store-bought seafood in Australia that exhibited far more MP contamination than wild-caught seafood. At least in one case, this was attributed to plastic cutting boards [36]. In contrast to microplastic ingested by organisms such as by fish, much of the microplastic issuing from the cutting boards lodges in the meat is often overlooked, cannot be easily removed from the produce by washing, and may change its properties during the cooking process [35]. The current contribution widens our previous findings on MP contamination of shop-bought produce to encompass fish and chicken meat bought in markets both in the United Arab Emirates and Kuwait.

## 2. Materials and Methods

***Acquisition of the food samples and cutting boards***. Specimen of six species of fish—giant trevally [*Caranx ignobilis* (Forsskål 1775)], Nile tilapia [*Oreochromis nilotica* (Linnaeus 1758)], threadfin bream [*Nemipterus bipunctatus* (Valenciennes 1830)], mackerel scad [*Decapterus macarellus* (Cuvier 1833)], orange spotted trevally [*Carangoides bajad* (Fabricius 1775)] and king mackerel [*Scomberomorus cavalla* (Cuvier 1829)] were bought at a fish market in the United Arab Emirates. Cut chicken samples were acquired from butchers and two chain supermarkets in the United Arab Emirates. In addition, cut salmon fillet [*Salmo salar* (Linnaeus 1758)] and cut chicken fillet were purchased from a Kuwaiti market. All fish and chicken samples were cut on plastic cutting boards by the vendors. It must be noted that all analyzed samples were cut in the shops as they would be for typical consumers. *No* further cutting of these samples was carried out in our laboratory. In addition, we asked that one sample was not cut but shredded in a meat grinder (SAP, Italy) to ascertain the origin of plastic contamination in the food samples that we analyzed. A further sample was cut in the shop on a wooden bamboo board. Neither of these two samples showed any MP contamination when analyzed. The chicken and fish samples were cut in our presence so that the type of plastic board used was known to us in all but one case. Therefore, it was possible for us to buy cutting boards of the same type from the shops that supplied the fish markets and supermarkets with the cutting boards. These cutting boards [Figure 1; (a) CB-1 white; (b) CB-2 red; (c) CB-3 green; (d) CB-4 yellow] were used for mechanical tests to measure their tensile strength, hardness and wear resistance to better understand their material integrity. 

***Contamination prevention*.** All the glassware used was washed with distilled water and dried in an oven (Ecocell, MMM group). Cotton lab coats, nitrile gloves, and face masks were worn at all times. Ziploc bags were tested for microplastic particles before use. All surfaces were cleaned and inspected before the experiments. Filter papers were inspected under the microscope before use. Blank samples were performed in parallel. 

***Digestion***. The digestion follows a published procedure [35]. In a single necked flask (500 mL) with a stopper, the (chicken/fish) meat sample (8–12.5 g) was added to an aqueous KOH solution (1 g KOH in 10 mL distilled H_2_O per 1 g of meat), and the mixture was stirred at a fixed temperature of 75 °C for 10 h. Fish samples (4–36.5 g) were digested analogously. The cooled solution was filtered through a filter paper (Schleich and Schueller 1573, pore size 2.5 μm). The filtered material was washed with distilled water (for 15 g of original sample, 3 × 100 mL) and thereafter with CHCl_3_/CH_2_Cl_2_ (for 15 g original sample, 3 × 50 mL). The residue on the filter was air-dried, investigated under an optical microscope (Leica Zoom 2000, Leica Microsystems, Wetzlar, Germany), transferred to a weighing paper, weighed and analyzed under a stereoscope (Olympus Model SZ2-ILST, Shinjuku, Japan). The filtrate was filtered through a second filter paper (Schleicher and Schuell 589/3, Dassel, Germany). The second filtration should allow for the capture of MP with a size as low as 6 μm, but no solid particles were found in the second filtration process. The chicken meat and fish digestion and analysis were performed in triplicate for each sample (C1–C6 and F1–F6).

For digestion of the chicken and fish samples, potassium hydroxide (KOH, Emsure^®^, Merck, Darmstadt, Germany), dichloromethane (CH_2_Cl_2_, Aldrich, St. Louis, MO, USA) and chloroform (CHCl_3_, Aldrich) were used without purification.

***Washing and grilling***. A king mackerel (*Scomberomorus cavalla*) specimen was washed under running cold water for 1 min. and dried in open air. Three samples of the washed king mackerel were taken and digested as detailed above. For the grilled king mackerel (*Scomberomorus cavalla*) specimen, a separate sample (45.94 g) was grilled in a Crown i-fryer (Model CL-113, 1500 W) at 200 °C for 15 min. Thereafter, the sample was cooled to room temperature and examined directly under a stereoscope (Model SZ2-2ILST).

***Analysis of the plastic particles***. The microplastic particles were weighed with an electronic balance either of type Radwag AS220.R2 (readability limit: 0.1 mg) or of type KERN ABS220-4N (readability limit: 0.1 mg). Photomicrographs of the plastic particles were taken with a stereoscope (Model SZ2-ILST). These were analyzed with Fiji Image J software for size and shape [37,38], including the Feret’s diameter. The Feret’s diameter was used to measure the size of plastic particles. The Feret’s diameter corresponds to the longest distance between any two points along the particle boundary [39]. Fourier Transform Infrared (FT-IR) spectroscopy was carried out on a Perkin Elmer Spectrum 2 FT-IR spectrometer as KBr pellets to determine the composition of the plastic particles found in the chicken meat and fish. The particles were found to be made of polythene (see also below).

***Analysis of the plastic cutting boards***. A part of the cutting board was ablated by filing and Fourier Transform Infrared (FT-IR) spectroscopy on the filed plastic particles was carried out with a Perkin Elmer Spectrum 2 FT-IR spectrometer. The cutting boards were found to be made of polythene with the following characteristic bands [40]: 2923 (asymmetric stretching CH_2_), 2852 (symmetric stretching CH_2_), 1671 (CH_2_-deformation), 1384 (symmetric terminal CH_3_ deformation), 729 and 719 cm^−1^ (CH_2_-rocking). Differential scanning calorimetry (DSC-60 Plus Differential Scanning Calorimeter, calorimetric measurement range ±150 mW, measurement performed under N_2_, Al crimp pans with lid, control software Thermal analysis workstation TA-60WS, range 23–250 °C [CB-3] and 23–400 °C [CB-4]) was carried out for CB-3 and CB-4. The density of the cutting boards CB-1–CB-4 was measured by displacement technique and was found to be 0.89–0.94 g/cm^3^.

***Mechanical analysis of the cutting boards***. To understand the structural integrity and performance of the materials, hardness, tensile and wear tests were conducted with the plastic cutting boards. The *hardness tests* were conducted according to ASTM D2240 standard using a Shore-D durometer. These tests are used to determine a material’s resistance to surface indentation [41]. The *tensile tests* were conducted on the plastic cutting boards to determine the highest load value that the material can bear before failure [42,43]. The *tensile tests* were conducted according to the ASTM D638 standard using the MTS universal testing machine (100 kN maximum capacity). The thickness of the cutting boards was 20 mm each and was machined to 14 mm for preparing the Type-III flat tensile specimens as shown in Figure 1a,b. Figure 1c,d shows the fractured specimens after the tensile tests. Five specimens were made from each cutting board using CNC machining. The tests were conducted at a crosshead speed of 50 mm/min. The *wear tests* were conducted using a pin-on-disk apparatus according to ASTM G99 standard. Three cylindrical pin specimens of 30 mm in length and 10 mm in diameter were prepared from each cutting board. The disc material is EN31 (hardened to 58 to 62 HRC). The following test conditions were used along with dry sliding: 30 mm track diameter, 300 rpm sliding speed, 1000 m sliding distance, and 10 N normal force. Moreover, the surface of the pin specimen was smoothed to 600-grit emery paper.

## 3. Results

All five chicken samples, cut from whole chicken which included bone, were bought in different markets in the United Arab Emirates (UAE) and a chicken fillet sample bought in a Kuwaiti market displayed MP contamination. The chicken from the markets of UAE was cut up on four types of plastic cutting boards, each type exhibiting a different color: white, yellow, green and red. From their color and from the fact that MP particles were shown to be made of polyethene just as the cutting boards it could be inferred that the MP contamination stemmed from the plastic cutting boards the chicken was cut upon (Figure 2, Table 1). Each cut sample (C1–C6) was analyzed in triplicate, and only for the chicken fillet sample were two of the three tests devoid of MP. Otherwise, the highest mass of MP was found in chicken sample 2 (C2, 0.29 ± 0.25 mg/g meat) whereas, the lowest weight was found for C6 (0.066 ± 0.025 mg/g meat). Sample 3 had the highest number (1.19 ± 0.72 particles) of MP particles per gram of meat of all chicken samples, whereas C6 (see also above) had the lowest number (0.03 ± 0.04 particles). Overall, per gram of cut chicken sample, the most MP particles stemmed from white cutting boards (1.2 ± 0.74 particles) followed by red (0.9 ± 0.36 particles), yellow and lastly green plastic cutting boards (0.56 ± 0.39 particles).

Five of the six studied cut fish samples had MP content. The size range of the extracted MP was found to be between 15.6–1151.1 μm, with an average size of 129.1 ± 138.2 μm (Table 2)**.** Sample F5 (orange-spotted trevally, *Carangoides bajad*) was found to have the highest number of MP (2.6 ± 2.8 particles/g fish). No MP particles were found in sample F4 (mackerel scad, *Decapterus macarellus*), bought in a UAE market. The average weight of MP of all fish samples, in which MP was found, was 0.0942 ± 0.1 mg/g fish.

The effect of washing the fish before food preparation on MP removal from the fish was investigated by washing a sample of king mackerel (F6, *Scomberomorus cavalla*) for 1 min under tap water, using approximately 5 L of water. It was found that washing could reduce the MP content in the fish meat from 1.17 ± 0.57 MP/g to 0.03 ± 0.03 MP/g. 

A sample of king mackerel (F6, *Scomberomorus cavalla*) was grilled in a Crown i-fryer (Model CL-113, 1500 W) at 200 °C for 15 min. It was found that the MP particles melted during the grilling process. Neighboring MP particles partially fused in their molten state. Moreover, MP particles were found to have changed their shape, where the most notable form after the grilling process had been completed was that of a spherical bubble. Due to the partial merging of the particles, the average size of the microplastics in the grilled fish with 316.7 ± 116.4 µm was found to be larger than the average size (192.0 ± 89.1 µm) of MP in the fish meat before grilling (Figure 3).

When studying the plastic cutting boards that are used in the markets, it was seen by DSC analysis [44,45] that the melting temperature of all cutting boards (CB-1–CB-4) was below 115 °C [CB-1 onset 104.6 °C; peak 111.5 °C), CB-2 (onset 98.3 °C, peak 112 °C), CB-3 (onset 102.0 °C; peak 111.1 °C) and CB-4 (onset 101.1 °C; peak 112.3 °C) (Figure 4). This indicates the material to be low-density polythene (LDPE) [46] (Figure 4). These runs conform with the density measurement of the cutting board material (<1 g/cm^3^). Furthermore, the DSC results of CB-1–CB-4 correlate with the values for hardness, wear and tensile tests (Figure 5 and Figure 6). The hardness values of CB-1 and CB-2 were given previously, the hardness values for CB-4 (yellow cutting board) varied from 50 to 54 with an average value of 51.9, while the hardness values of CB-3 (green cutting board) varied from 49 to 54 with an average value of 52.5. Figure 6 shows the tensile stress–strain curves for the materials. As most of the extension is happening in the narrow section of the tensile specimen, the strains are calculated by dividing the extension (length after/length before) between the grips within the narrow section (57 mm). The first peak stress (tensile strength/yield strength) varied from 10.7 to 11.1 MPa with a mean value of 10.9 MPa for CB-3 (green cutting board), while the tensile strength varied from 10.6 to 10.9 MPa with a mean value of 10.8 MPa for CB-4 (yellow cutting board). The tensile strengths of CB-3 and CB-4 were found to be very close in value, however, CB-4 shows more elongation than CB-3. This is obvious from the stress–strain curves (Figure 6) and from the fractured specimens (Figure 5). The recorded average wear rate for CB-3 (green cutting board) was 0.393 mm^3^, while the recorded average wear rate for CB-4 (yellow cutting board) was 0.284 mm^3^.

## 4. Discussion

For some time, wooden cutting boards have been replaced in part in food production sites, at least in meat and poultry processing settings, by cutting boards made of other materials, such as of high-density polythene (HDPE) or low-density polythene (LDPE) [47]. This is mainly due to the worry that microorganisms could penetrate and be trapped in the pores of wooden boards, although this has been heavily debated by a number of researchers [48]. Nevertheless, the Food Safety and Inspection Service of the US Department of Agriculture [49] states that one can choose between a wooden cutting board and a cutting board with a non-porous surface made from plastic, marble or pyroceramic without the worry of impacting one’s health. It also states that plastic and wooden boards wear out over time and should be replaced. The wearing out of the plastic boards is due to the flaking off of plastic pieces from the cutting board, as described above, which can at the very end affect the structural integrity of the cutting board itself. Previously, the authors noted that typically as much as 875 g polythene could be lost at the end of a cutting board’s lifetime, in the case of cutting goat meat, that equates to 400 kg of meat cut on the board [35]. It must be noted, however, that the mechanical tests showed that the plastic cutting boards are quite durable, with a low wear rate of 0.284–0.393 mm^3^.

In the natural environment, MP ingestion of fish is influenced by the location but can also be a function of fish species, where certain fish species such as the bouge fish (*Boops boops*) in the Mediterranean Sea [50] have already been used as a bio-indicator for the presence and location of MPs in the aquatic environment. In the present case, the MP contamination is not a function of the specific fish species considered. However, the fish species chosen for the study are all large, strong fish, and some force is needed to cut them into smaller pieces. This can be comprehended when comparing the MP contamination of samples cut on a plastic cutting board from a fillet that itself has been prepared on a surface other than plastic (F7). Cutting a fillet into smaller pieces necessitates less force and therefore, the MP load is less.

We observed earlier that MPs as contamination in food undergo a physical transformation at the high temperatures of the food preparation processes due to the relatively low melting point of MPs [35]. This is especially true for polythene-derived MPs, which melt during frying or pressure cooking processes or when the food is grilled as shown in the current paper. Upon cooling, the MPs can crystallize but can also stabilize in an amorphous state. The authors have seen examples, where polythene MPs come to reside in an amorphous state upon cooling and where small polythene crystallites grow on the main body of the solidified MP. During the cooling process, small molecules such as fats can be entrained in the hardened MPs. In the future, it may also be of interest to study whether MPs as contaminants in food release additives into the food during the cooking process.

As noted in our earlier paper [35], not all MP generated in the cutting process ends up in the cut fish. About 50% remains on the cutting board and will be washed into the sink. Thus, at the market, extensive washing of the cutting board prior to cutting was seen especially for the king mackerel and giant trevally, both of which still exhibited significant MP contamination. In the case of 1 kg king mackerel cut into fine pieces, one could expect that as much as 2.47 g of plastic could find its way into the wastewater. This can be compared with 33 mg MP released into the sink for one wash with a typical MP containing rinse-off cosmetic [51,52]. From this point of view, MP released into the wastewater from cutting food on plastic cutting boards can be seen as a major contributor of MP in wastewater.

The problem of MP contamination of food through plastic cutting boards is a global phenomenon, where the authors have collected evidence from markets of other countries as from the South Asian region. That Kuwait and the United Arab Emirates were the countries chosen for this study was incidental as it is where the authors currently reside. To the best of our knowledge, apart from our previous communication of MP contamination in goat meat and beef, this presents the first published study of MP contamination in food produce stemming from plastic cutting boards. When comparing the highest observed MP contamination in chicken (0.29 ± 0.25 mg MP/g chicken) it is lower than the previously observed contamination in goat meat and beef (1.64 ± 0.46 mg MP/g meat). The highest MP contamination overall was found in king mackerel (2.47 ± 0.4 mg MP/g fish).

Microplastic contamination in cut produce can be avoided by using wooden cutting boards. This was also shown in this study where fish and chicken cut in the shops on bamboo boards did not contain microplastic at all. Meanwhile, steel and marble cutting boards are being re-introduced in the markets, too. In all cases, cutting boards have a certain lifetime, where more fragmentation of a board is found near the end of its lifetime. Therefore, cutting boards should be replaced in time. This is especially true for plastic and wooden boards.

## 5. Conclusions

Plastic cutting boards were found to be the source of microplastic (MP) contamination in raw cut fish and chicken from Middle Eastern markets with as much as 2.47 ± 0.4 mg MP/g fish and 0.29 ± 0.25 mg MP/g chicken. Only one of seven fish specimens and none of the chicken specimens were found to be devoid of MP contamination. Washing the raw food produce reduced the MP contamination, for instance from 1.17 ± 0.57 MP/g fish to 0.03 ± 0.03 MP/g fish but did not remove it completely. Food preparation involving heat induces the MP particles to melt, where upon cooling, the particles solidify in new forms, sometimes entraining material from the fish tissues. Not all MPs released from the plastic cutting boards during the cutting process remain on the produce, but it is believed that as much as 50% of MPs generated find their way into the wash-water during the cleaning of the cutting boards. Thus, plastic cutting boards constitute a major source of MPs in wastewater.

## Figures and Tables

**Figure 1 ijerph-19-13442-f001:**
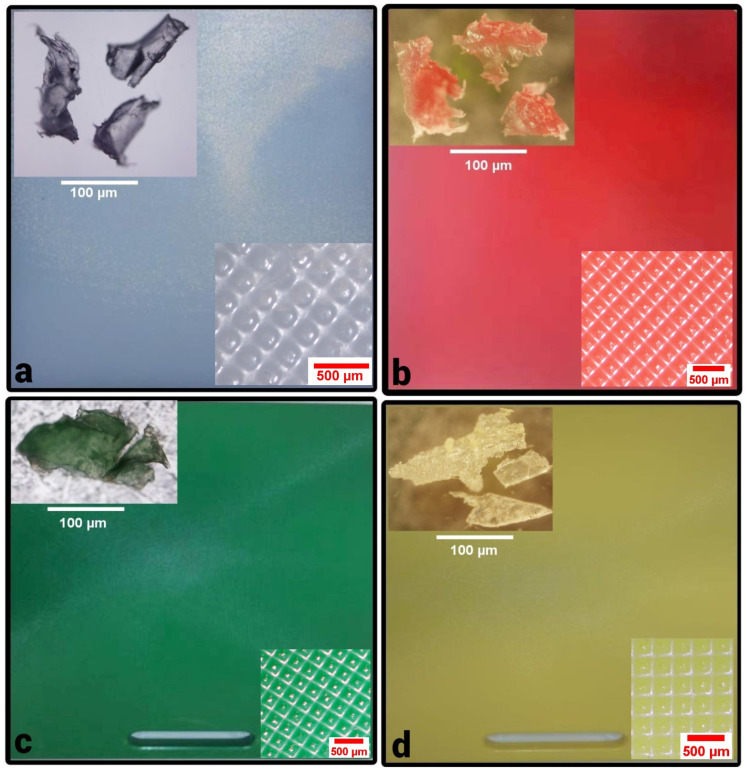
Examples of cutting boards used by markets with corresponding generated microplastic fragments (upper left corner, photos taken with microscope Olympus model BX41TF) and surface texture of the cutting boards (lower right corner, photos taken with stereoscope model SZ2-ILST): (**a**) CB-1 (white/colorless); CB-2 (red); CB-3 (green); CB-4 (yellow) where are (**b**–**d**).

**Figure 2 ijerph-19-13442-f002:**
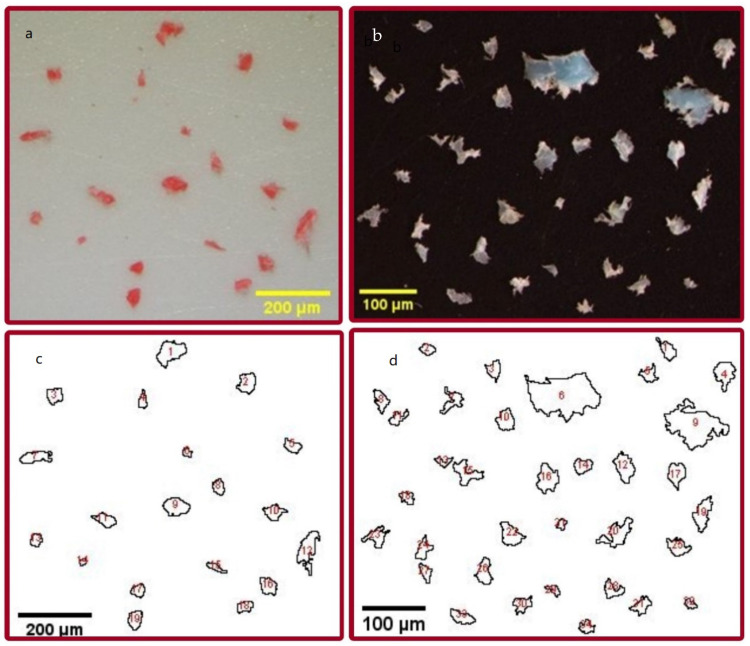
Original photos (**a**,**b**) and photos processed with Image J (**c**,**d**) of MP obtained through the digestion of (**a**,**c**) chicken sample 3 (C3) and (**b**,**d**) orange-spotted trevally (F5).

**Figure 3 ijerph-19-13442-f003:**
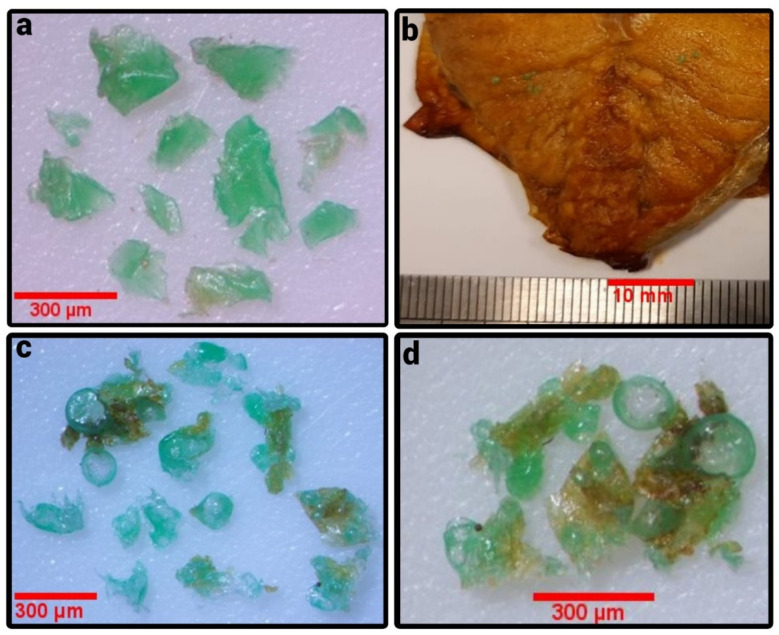
(**a**) MP stemming from cut raw king mackerel (*Scomberomorus cavalla*) after digestion with aq. KOH; (**b**) part of a grilled king mackerel piece with adhering MPs; (**c**,**d**) MPs isolated from the grilled king mackerel.

**Figure 4 ijerph-19-13442-f004:**
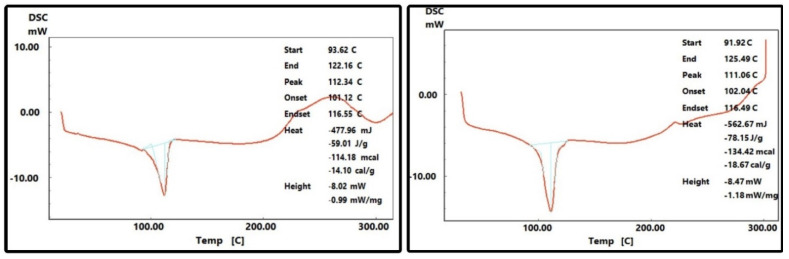
Differential scanning calorimetry (DSC) curves of polythene material from CB-3 (**right**) and CB-4 (**left**).

**Figure 5 ijerph-19-13442-f005:**
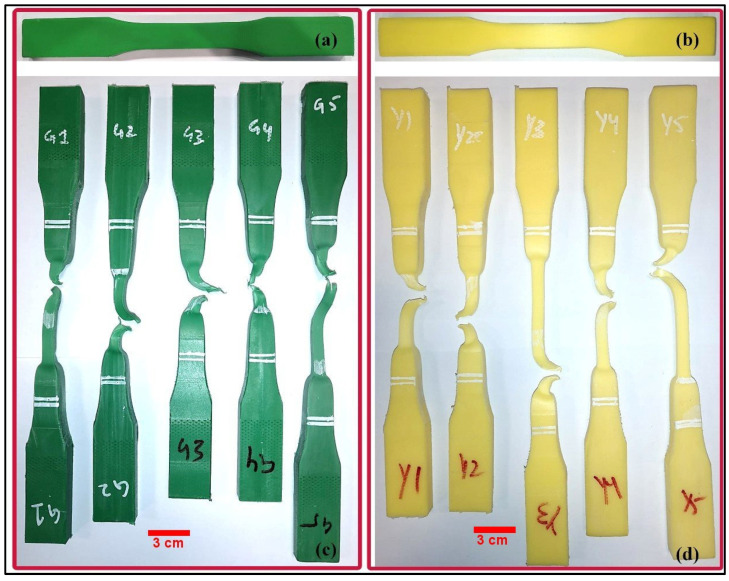
Prepared tensile specimens of CB-3 (**a**); CB-4 (**b**); fractured tensile specimens of CB-3 (**c**); and CB-4 (**d**).

**Figure 6 ijerph-19-13442-f006:**
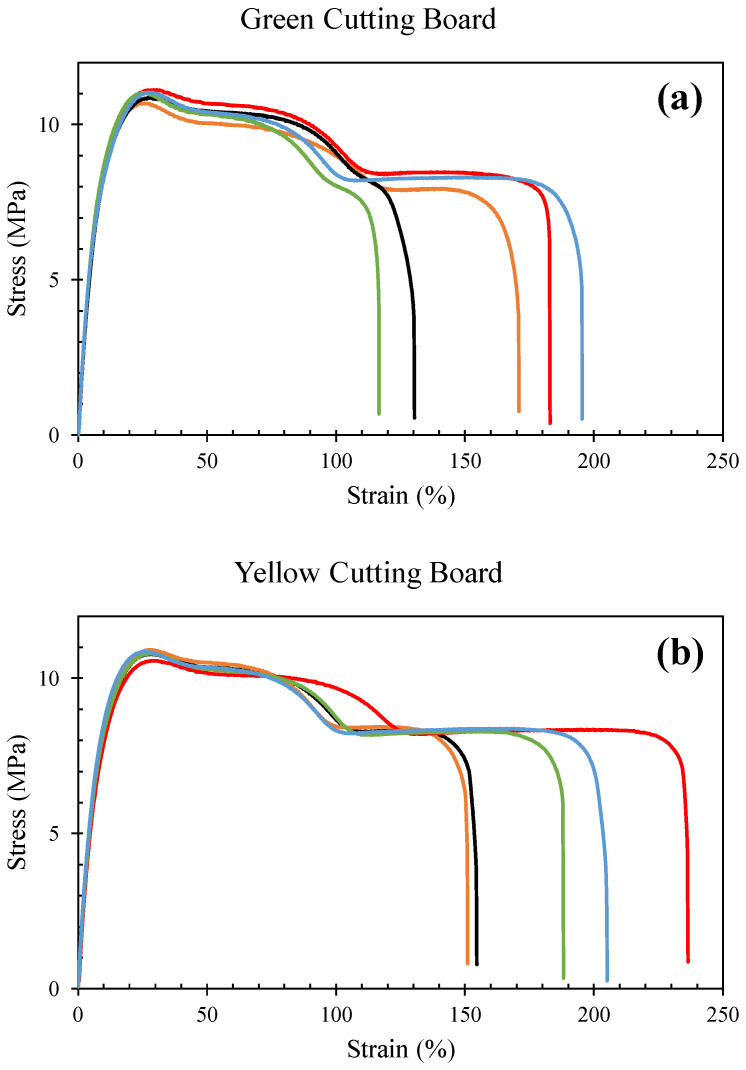
Stress-strain curves of the plastic cutting boards: (**a**) green cutting board (CB-3); and (**b**) yellow cutting board (CB-4). Five tensile specimens were prepared from each cutting board to determine the mean tensile strength of the cutting boards. The different colored lines show the stress-strain curves obtained from tensile tests of the five specimens. It can be observed from the figure that very consistent tensile strength values were obtained for all the five specimens.

**Table 1 ijerph-19-13442-t001:** Mass, number, size distribution and color of MPs in chicken meat samples.

Sample Name	Wt. (mg) of MPs/g of Food ± SD	No. of MPs Pieces/g of Food ± SD	MPs Size Range (μm)	Mean Size (µm) of MPs ± SD	Color of MPs	Color of CB Used
C1 (UAE)	0.22 ± 0.11	1.1 ± 0.76	59.1–249.7	117.2 ± 42.5	white/colorless	white
C2 (UAE)	0.29 ± 0.25	1.19 ± 0.72	23.7–1454.5	148.6 ± 97	white/colorless	white
C3 (UAE)	0.147 ± 0.06	0.9 ± 0.36	8.24–984.2	95.1 ± 215.1	red	red
C4 (UAE)	0.11 ± 0.031	0.63 ± 0.4	33.9–462.3	178.3 ± 143.6	yellow	yellow
C5 (UAE)	0.066 ± 0.025	0.52 ± 0.38	14.3–152.7	74.6 ± 7.8	yellow	yellow
C6 (Kuwait)	0.01 ± 0.02	0.03 ± 0.04	34.1	11.35 ± 19.7	green	green

The size range of MP found in the chicken samples was between 8.24–1454.5 μm, with an average size of 104.2 ± 84.33 μm.

**Table 2 ijerph-19-13442-t002:** Size distribution and characterization of MPs in fish samples.

Sample Name	Fish	Wt. (mg) of MPs/g of Food ± SD	No. of MPs Pieces/g of Food ± SD	MPs Size Range (μm)	Mean Size (µm) of MPs ± SD	Color of MPs	Color of CB Used
F1 (UAE)	Giant trevally (*Caranx ignobilis*)	0.074 ± 0.013	0.024 ± 0.42	62.1–199.5	136.4 ± 69.4	red	red
F2 (UAE)	Nile tilapia (*Tilapia nilotica*)	0.0041 ± 0.0071	0.014 ± 0.024	144.47	48.2 ± 83.4	green	green
F3 (UAE)	Threadfin bream (*Nemipterus bipunctatus*)	0.31 ± 0.47	0.52 ± 0.22	15.6–94.6	52.69 ± 20.6	green	green
F4 (UAE)	Mackarel scad (*Decapterus macarellus*)	0	0	0	0	yellow	yellow
F5 (UAE)	Orange-spotted trevally (*Carangoides bajad*)	0.097 ± 0.09	2.60 ± 2.80	16.6–146.3	49.4 ± 8.72	blue	blue
F6 (UAE)	King mackerel (*Scomberomorus cavalla*)	2.47 ± 0.4	1.17 ± 0.57	51.8–503.5	192.0 ± 89.1	green	green
F6A (UAE)	King mackerel (*Scomberomorus cavalla*) washed	0.0022 ± 0.0021	0.03 ± 0.03	89.7–737	309.7 ± 335.3	green	green
F7 (Kuwait)	Atlantic salmon (*Salmo salar*)	0.08 ± 0.01	0.16 ± 0.06	67.7–1151.1	544.3 ± 459.8	white	white

## Data Availability

Further data can be obtained from the corresponding author.

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
