# Peer review of "Microplastic Contamination of Chicken Meat and Fish through Plastic Cutting Boards"

_ijerph, 2022, doi:10.3390/ijerph192013442_

Round 1

Reviewer 1 Report

The authors should provide more explanations on how the work on the blank samples was done and whether microplastics were observed in the blank samples.

In the discussion section, the results of the work should be compared with the work of others and the reasons should be mentioned.

- The phrase of " Please check the caption" should be removed from some titles of the figures.

Author Response

The authors thank the reviewer for his/her insightful comment and his/her valuable time.

To point 1: The actual cutting of the chicken/fish samples was exclusively carried out in the shops/supermarkets. Except for in one case, the authors have seen the cutting boards the chicken/fish was cut on as these were cut in open view. Thus the colors of the boards are known to us. The shop-keepers told us where the cutting boards were bought and we bought the same boards from the same shop to perform the mechanical tests on the boards. The color and texture of plastic fragments from the bought plastic cutting boards matched the microplastic particles found in the bought produce.

Also, we had asked two shops to cut samples on wooden (bamboo) cutting boards and also to mince samples through a grinder to ascertain that the source of the plastic in the food was indeed from the plastic cutting boards. No microplastics were found in these samples.  We have added lines 75-76 and 79-81 to make this clearer.

To point 2: 

To the best of our knowledge, no actual study has been done on plastic cutting boards as a source of microplastics in food. We have published a previous paper on contamination of goat meat and beef (https://doi.org/10.1080/19440049.2021.2017002), and the main results of that paper are already mentioned in the text and compared to the current results. We have added the lines 294-300 to discuss this point.

To point 3:

We have removed the phrase "please check the caption: from some titles of the figures.

Reviewer 2 Report

This manuscript study the microplastic contamination of chicken meat and fish through plastic cutting boards.

1. How do you distinguish the microplastic from the plastic cutting boards or other sources?

2. The experimental design looks not very clear, you cut the chicken meat and fish from 4 different  plastic cutting boards? You,d better give us a table.

3. The introductiong should list the objects of this study.

Author Response

The authors thank the reviewer for his/her insightful comment and his/her valuable time.

To point 1: The actual cutting of the chicken/fish samples was exclusively carried out in the shops/supermarkets. Except for in one case, the authors have seen the cutting boards the chicken/fish was cut on as these were cut in open view. Thus the colors of the boards are known to us. The shop-keepers told us where the cutting boards were bought and we bought the same boards from the same shop to perform the mechanical tests on the boards. The color and texture of plastic fragments from the bought plastic cutting boards matched the microplastic particles found in the bought produce.

Also, we had asked two shops to cut samples on wooden (bamboo) cutting boards and also to mince samples through a grinder to ascertain that the source of the plastic in the food was indeed from the plastic cutting boards. No microplastics were found in these samples. We have added lines 75-77 and 79-81 to discuss this point.

To point 2: 

We have incorporated the information of the colors of the cutting boards that the shops used in the Tables 1 and 2. The information on the colors of the cutting boards is also summarized below.

Sample name                Color of Cutting board

C1 (UAE)                                  white/colorless

C2 (UAE)                                  white/colorless

C3 (UAE)                                  red

C4 (UAE)                                 yellow

C5 (UAE)                                 yellow

C6 (Kuwait)                             green

F1 (UAE)                                  red

F2 (UAE)                                  green

F3 (UAE)                                  green

F4 (UAE)                                  yellow

F5 (UAE)                                  blue

F6 (UAE)                                 green

F6A (UAE)                              green

F7 (Kuwait)                            white

3. The introduction should list the objects of this study.

This study was carried out to investigate possibility that plastic cutting boards could lead to microplastic contamination in fish and chicken. We have added the objective to the introduction of this study (lines 63-65).

Reviewer 3 Report

Dear Authors,
The manuscript is interesting, but should be improved showing the novelty of your work - especially in the Introduction and Discussion section. In addition, the manuscript may be presented as a Short Communication. Please find below specific suggestions.

Keywords
Please avoid words used in the title. For instance, "Fish".   1. Introduction
Line 28-30. Please consider references.

Line 32-34. Please consider references.

Line 51. "liver, spleen, and kidneys." Please consider references.

General comment. ​Chicken seems to be overlooked in the text. Please include a paragraph about novelty for the study of chicken.

2. Materials and Methods
Line 68-71. Please include the author of the species. For instance, Caranx ignobilis (Forsskål 1775).

Line 68-71. Please revise the validity of all species. You may check this in https://researcharchive.calacademy.org/research/ichthyology/catalog/fishcatmain.asp

General comment. ​Do you analyse fish gut?

4. Discussion
Line 266-275. Please include references to support the sentences of this paragraph.

General comment. Please include a general paragraph with the main findings.

General comment. Please discuss the novelty of your study.

General comment. Please discuss ways to combat the problem of contamination in the markets.

Author Response

We thank the reviewer for the comments and the valuable time spent on this manuscript.

To point 1:

We have heeded your advice and have replaced the words "Chicken and "Fish" with "Food Preparation"

To point 2: 

We have added reference 8 and renumbered the remaining references.

To point 3:

we have deleted part of the sentence (kidneys,,,,). The reason for this is that the ACS gave a pre-press conference at the ACS Fall Meeting 2022, where the findings of the group of Prof. Halden on MP content in human kidney and other organs were disseminated, but apparently the paper was not given at the conference.

Point 4: Chicken is discussed in lines 163 - 178.

Point 5: Thank you for your comment and pointing out the website - we have added the author of the species.

General comment:

Here, the shops cut the gut away, when they prepare the fish for you. However, in a separate study we are analyzing 80 whole fish samples bought at local markets for MP content in the gut (we are about half way through). The fish come from the Emirati Arabian Gulf coast.   

Point 6: We have added the reference for lines 266-276 (line numbers for the original draft).

Point 7: We have added lines 294-296 to discuss the novelty of the study.

Point 8: We have added lines 301-307 to discuss how to combat the problem of MP contamination of the food during the food processing.